# Ultrastructure of Immatures Stages and Life Cycle of *Helicobia aurescens* (Diptera: Sarcophagidae: Sarcophaginae)

**DOI:** 10.3390/insects15100753

**Published:** 2024-09-28

**Authors:** Lucas Barbosa Cortinhas, Paloma Martins Mendonça, Eliane Gomes Perrut, Rodrigo Rocha Barbosa, Jacenir Reis dos Santos-Mallet, Margareth Maria de Carvalho Queiroz

**Affiliations:** 1Laboratório Integrado, Simulídeos e Oncocercose & Entomologia Médica e Forense, Instituto Oswaldo Cruz, Fundação Oswaldo Cruz, Rio de Janeiro 21040-360, RJ, Brazil; lucas.cortinhas@gmail.com (L.B.C.); elianegomes74ster@gmail.com (E.G.P.); mmcqueiroz@ioc.fiocruz.br (M.M.d.C.Q.); 2Programa de Pós-Graduação em Biodiversidade e Saúde Instituto Oswaldo Cruz, Fundação Oswaldo Cruz, Rio de Janeiro 21040-360, RJ, Brazil; 3Mestrado Profissional em Ciências Ambientais, Universidade de Vassouras, Vassouras 27700-000, RJ, Brazil; 4UniFoa—Centro Universitário de Volta Redonda, Campus Volta Redonda, Volta Redonda 27215-630, RJ, Brazil; rodrigorb1@hotmail.com; 5Laboratório Interdisciplinar de Vigilância Entomológica em Diptera e Hemiptera, Instituto Oswaldo Cruz, Fundação Oswaldo Cruz, Rio de Janeiro 21040-360, RJ, Brazil; jacenir.mallet@fiocruz.br; 6Laboratório de Vigilância e Biodiversidade em Saúde, Universidade Iguaçu, Nova Iguaçu 26260-045, RJ, Brazil

**Keywords:** scanning electron microscopy, bionomy, flesh fly, morphology, biology, development, medical and forensic entomology

## Abstract

**Simple Summary:**

Flies are often associated with urban waste, myiasis and decomposing bodies. *Helicobia aurescens* is found in carcasses and could be useful to forensic entomologists in the determination of the postmortem interval. However, a few studies are available regarding the morphology and biology of this species. We describe aspects of the life cycle at two temperatures (27 ± 1 °C and 29 ± 1 °C) and analyze the morphological characteristics of the eggs, larvae, and puparia of *H. aurescens* using scanning electron microscopy (SEM). A lower temperature affected the flesh fly development, increasing the total development time. The immature morphology was very similar to those of the others sarcophagids. No hatching lines or median areas were detected on the eggshells. The first-instar larva is very similar to those of the other species. The anterior spiracles have six or seven ramifications aligned regularly at the second instar. But, on the third instar, these structures have eight ramifications in a regular row and are located dorsolaterally. The puparia are similar to that of the third-instar larvae.

**Abstract:**

*Helicobia aurescens* is a flesh fly associated with pig and rat carcasses. This study aims to describe the life cycle at two temperatures (27 ± 1 °C and 29 ± 1 °C) and analyze the morphological characteristics of the eggs, larvae, and puparia of *H. aurescens* using scanning electron microscopy (SEM). Temperature is an abiotic factor that greatly influences the development of insects. The larval development of *H. aurescens* lasts longer at 27 ± 1 °C than it does at 29 ± 1 °C, affecting the growth of newly hatched larvae into adults. The females larviposited three times more at 27 °C than they did at 29 °C, and the number of days laying larvae was also greater. At 27 °C, they laid larvae for 38 days, whereas, at 29 °C, the females larviposited for 21 days. No hatching lines or median areas were detected on the exochorion with SEM, as in the other sarcophagid species. The first-instar larva is very similar to those of the other species. The second instar has anterior spiracles present on the first thoracic segment, with six or seven ramifications aligned regularly. On the third instar, these structures have eight ramifications in a regular row and are located dorsolaterally. The puparium morphology is similar to that of the third-instar larvae.

## 1. Introduction

The family Sarcophagidae, commonly referred as flesh flies, is distributed worldwide, comprises around 355 genera and 3000 species, and consists of the subfamilies Miltogramminae, Paramacronychiinae, and Sarcophaginae [1]. Present mainly in the neotropics, Sarcophaginae has the highest diversity of all the subfamilies within the family, with approximately 780 species described in the region [2].

The taxonomic descriptions of sarcophagid species are primarily based on the characteristics of male genitalia. Nonetheless, it is the females that search for protein sources to develop their ovarioles and larviposition. Additionally, the most important stage in public health and forensic entomology is the immature one. Sarcophagids show a variety of behaviors, with the larvae exhibiting the highest diversity [2]. This stage may exhibit different feeding habits that include feeding on decaying organic matter (saprophagous), dead animal (vertebrate, insects, and snails), carcasses (necrophagous), or excrement (coprophagous) [3,4]. Certain species are parasitic, either obligately or facultatively, or even kleptoparasites of bees and wasps, while a small number of species produce myiasis-causing larvae [5,6].

*Helicobia* Coquillett, 1895, is a large genus originally from North and South America and includes 35 valid species, most of which are classified as Neotropical species [2]. Initially, the *Helicobia* species was collected from dead or injured insects and snails [7,8]. However, more recently, *Helicobia aurescens* Townsend, 1927, was found in association with pig, dog, and rat carcasses, bovine livers, and fishes [9,10,11,12].

The taxonomic position of *Helicobia* remains a topic of discussion [2,13], being one of the most recent phylogenies for the group based on the male terminalia functionality [14]. To confirm its status, additional studies on the species’ biology and immature morphology are necessary, as the genitalia characteristics of males alone do not provide sufficient evidence for a definitive consensus. In addition to the lack of morphological information and the diversity of the habits, many biological aspects of *Helicobia* species are still unknown, and larval growth, for example, how a variation in temperature could affect the species. Grella [15] successfully collected *H. aurescens, H. pilifera* Lopes, 1939, and *H. pilipleura* Lopes, 1939, but just the last species was reared until the second generation in a laboratory, emphasizing the need for studies on this genus.

The knowledge of the family Sarcophagidae is mainly related to the morphology and biology of the adult stages, whilst there is a considerable lack of knowledge related to the immature stages, and when focused on Neotropical species, this lack of information is even greater. This study aims to describe the life cycle of *H. aurescens* at two different temperatures and characterize the morphological aspects of its eggs, larvae, and puparia utilizing scanning electron microscopy (SEM).

## 2. Materials and Methods

### 2.1. Fly Rearing

*Helicobia aurescens* specimens were collected at the Oswaldo Cruz Foundation campus in Manguinhos, Rio de Janeiro, RJ, using fish carcasses (sardines) as bait, which were cut into half, but both parts were used following the methodology described by Barbosa et al. [16].

The colonies were reared and maintained as described by Queiroz and Milward-de-Azevedo [17] at Laboratório Integrado: Simulídeos e Oncocercose & Entomologia Médica e Forense (LSOEMF), Oswaldo Cruz Institute, Oswaldo Cruz Foundation. The entire experiment was performed using second laboratory generation (F2).

### 2.2. Life Cycle

For the life cycle experiment, two hundred newly hatched larvae were collected from the species colony and divided into four groups of 50 individuals for each temperature. This low density of larvae prevented any measurable temperature increase caused by the heat generated by the larvae [18] from stimulating growth [19,20].

Each group was placed into a 100 mL plastic container with 50 g of putrefied meat and inside a 500 mL plastic container with vermiculite. As the third-instar larva began to wander, they were individually weighed using a high-precision weighing scale (Ohaus Pioneer Model PA 214CP) and stored in glass tubes containing vermiculite. Daily observations were made until the adults emerged, always at the same time.

The weight of mature larvae (L3), the duration of the larval (L1–L3) and pupal stages, the duration from L1 to the adult stage, and adult emergence were recorded for the bionomics data.

The pupation date, emergence of imagoes, and sex ratio were recorded, as well as any morphological anomalies in the adults. A bioassay was performed in a climatic chamber (Fanem Model 347CDG) using two different temperatures: 27 ± 1 °C and 29 ± 1 °C, and a relative humidity of 60 ± 10% and a 12 h light/dark cycle.

After adult emergence, 45 couples were transferred and maintained in three colony cages (15 couples per cage) with water and sugar ad libitum, and putrefied bovine meat was offered for larviposition. The longevity of the individuals and the number of larvae laid were recorded.

The Mann–Whitney test was used to evaluate longevity and larval weight. The differences in mortality between both temperatures were analyzed using chi-square (x^2^). Survivor curves are represented by the Weibull distribution [21].

### 2.3. Scanning Electron Microscopy (SEM)

A total of ten specimens of each developmental stage (eggs, first-, second-, and third-instar larvae, and puparia) were collected from the colony for the SEM experiment. The methodology for sacrifice, fixation, post-fixation, and dehydration has been described by Mendonça et al. [22]. The critical point drying method used super dry CO_2_ with Balzer’s apparatus [23]. The samples were then placed on metallic supports, coated with a thin layer of gold (20–30 nm), and examined under a Jeol JSM 6390LV scanning electron microscope (Akishima, Tokyo, Japan) of the Rudolf Barth Electron Microscopy Platform of the Instituto Oswaldo Cruz (FIOCRUZ). The terminology used to describe the morphology in this work followed Teskey [24].

## 3. Results and Discussion

### 3.1. Life Cycle

The larval development of *H. aurescens* lasts longer at 27 ± 1 °C than it does at 29 ± 1 °C (Table 1). The pupal stage and total development time were also longer at the lower temperature, with a statistical difference. It is well known that flesh flies can be used to estimate the postmortem interval (PMI), so it is important to know the accurate development time of each species and the time needed to complete each stage. In this paper, just the larval, pupal and complete developmental periods were analyzed.

Temperature is an abiotic factor that largely influences insect development, as it accelerates the metabolism, especially in the pupal stage [20,25]. Cunha [25] observed the effects of temperature on the intrapuparial development of *Peckia* (*Sarcodexia*) *lambens* (=*Sarcodexia lambens*) (Wiedemann, 1830), ranging from 7.5 days at 21 °C to 4 days at 31 °C. Grella [15] reported total development times of 14.5 days for *H. aurescens* at 25 ± 1 °C, 16 days for *H. pilifera*, and 15.33 days for *H. pilipleura* at the same temperature. This author [15] reported an insufficient amount of immature insects to create a growth curve, which is maybe because of the use of isolated females. The number of larvae successfully reared was not mentioned by the authors, but it could be inferred that the probable low larval density could have affected the developmental time of *H. aurescens*, motivating the larvae to stop eating the food prematurely. The ambient conditions and ground temperatures could directly affect larval development once the temperatures within the maggot feeding aggregations vary considerably, affecting the internal mass temperatures influenced directly by the larval density [26,27].

Viability analysis showed no statistical difference between the temperatures and developmental periods, except for the newly hatched larvae that grew into adults at 29 ± 1 °C. Only 67% of the larvae reached the adult stage (Table 1). da-Silva-Xavier et al. [28] also reported the lowest viability (67%) when rearing *Oxysarcodexia amorosa* (Schiner, 1868) at the same temperature mentioned above.

A statistically significant difference was observed in the weights of the mature larvae and the emergence of males and females in both temperature conditions, according to this study (Table 2). Nevertheless, Salviano et al. [29] reported that for *Peckia* (*Squamatodes*) *trivittata* (=*Squamatodes trivittata*) Curran, 1927, the females were lighter than the males at 27 °C (238 ± 34 mg and 257 ± 33 mg, respectively), suggesting sexual dimorphism in the body weight for this species.

Remaining in the larval stage for a prolonged period may be influenced by the low-level consumption of animal protein. In addition to affecting the duration of this period, a lack of animal protein may also affect the size and weight of both the larvae and adults. According to Slansky and Scriber [30], the larval weight could reflect the amount of energy and nutrients stored, which may directly affect the reproductive capacity. Amoudi et al. [31] studied the developmental time of *Sarcophaga* (*Liopygia*) *ruficornis* (Fabricius, 1794) at 12 constant temperatures ranging from 13 °C to 37 °C, indicating that the optimal temperature in terms of rapid development, a low mortality rate, and a higher weight was between 22 °C and 28 °C. As the temperature is specific for each species, more studies should be conducted to a better understanding of *H. aurescens* bionomy.

According to Table 2, the larvae abandoned the diet on the third, fourth, and fifth day after eclosion. At a temperature of 27 °C, there was no statistically significant difference in weight among the larvae that stopped eating the food on different days. However, at 29 °C, the larvae that abandoned the diet on the fifth day were the lightest, with a mean weight of 11.66 ± 3.28 mg. No information was found in the literature regarding the minimum larval weight for flesh fly larvae to become viable adults; instead, there were just studies about specific species [31]. This study showed that *H. aurescens* requires a minimum larval weight of 12mg to reach the adult stage at 27 °C, while a weight of 7mg was insufficient to complete development at 29 °C.

According to Parra [32], insect feeding could be retarded or inhibited simply by a deficit or surplus of essential nutrients. The larvae reared at 29 °C that abandoned the diet on the fifth day were lighter, which could be explained by the difficulty in obtaining the quantity and quality of nutrients needed for pupation (Table 2).

The life expectancy of males and females observed at both temperatures followed a Weibull distribution, as shown in Figure 1 and Figure 2. At both temperatures, the females lived longer than the males (27 °C: females = 37.9 days; males = 29 days; 29 °C: females = 21.8 days; males = 15.2 days). Mackerras [33] studied different species of flies and found that the females lived longer than the males. Ferraz [34] reported the same observation for the flesh flies *Peckia* (*Peckia*) *chrysostoma* (Wiedemann, 1830) and *Peckia* (*Peckia*) *ingens* (=*Adiscochaeta ingens*) (Walker, 1849). On the other hand, Oliveira et al. [35] reported a longer lifetime for the males of *Peckia* (*Pattonella*) *intermutans* (=*Pattonella intermutans*) (Walker, 1861). Furthermore, the females of *P*. (*S*.) *trivittata* lived longer than the males at 16 ± 1 °C and 50–60% RH; however, the males lived longer at 27 ± 1 °C and 70–80% RH [29].

Majumder et al. [36] reared adults of *Sarcophaga* (*Boettcherisca*) *peregrina* (=*Boettcherisca peregrina*) (Robineau-Desvoidy, 1830) at 25 ± 5 °C and 70–80% RH, both in pairs and unpaired conditions. One group was fed bovine liver, water, and sugar, while the other group was not provided with animal protein. These authors observed that the males lived longer in all the situations presented, but also concluded that the strong influence of the protein and paired conditions prolonged the lives of these flesh flies.

These observations highlight the absence of a developmental pattern among the species of the Sarcophagidae family. In this study, the insects lived longer at 27 °C compared to those at 29 °C, but there was no statistical difference between the groups. This is probably due to the higher temperatures accelerating their metabolism and reducing their longevity.

The females reared at 27 °C laid 1498 larvae and 303 eggs from day 10 to 51 of life, but only 11 eggs were viable. Larviposition reached the maximum on days 18, 26, and 35, with approximately 2.5 larvae per female. The larviposition rhythm at 27 °C is shown in Figure 3. At 29 °C, the females laid 396 larvae and 12 eggs from day 7 to day 28 of observation. The peak of larviposition occurred on days 12, 18, and 21 (Figure 4), with the same ratio of approximately 2.5 larvae per female. The females laid three times more larvae at 27 °C than those at 29 °C, and the number of days on which they laid larvae was higher. At 27 °C, larviposition occurred for 38 days, whereas, at 29 °C, the females larviposited for only 21 days.

Classically, sarcophagids are known to be larviparous, with the females laying first-instar larvae directly on a substrate. However, some authors have reported that flesh flies can lay eggs, as was observed in this study [37]. Zumpt [38] pointed out that oviparity and ovoviviparity can occur within the same flesh fly species, such as in *Sarcophaga* (*Bercaea*) *africa* (=*Bercaea africa*) (Wiedmann 1824). Sukontason et al. [39] described the eggshells of *Sarcophaga* (*Liosarcophaga*) *dux* (=*Liosarcophaga dux*) Thomson, 1869, and highlighted that the females can oviposit under optimal conditions, which can also be simulated under laboratory conditions.

### 3.2. Ultrastructure

#### 3.2.1. Eggs

The eggs have a bean-like shape (Figure 5A). The anterior end has a flattened apex, and the posterior end is domed. The median length is 0.89 ± 0.08 mm. No hatching line or median area was observed, so the dorsal and ventral surface location cannot be determined. Concave and convex surfaces were present (Figure 5B,C). The exochorion has a polygonal cell pattern that is more delimited (raised edges) at the anterior region and extends to the middle of the egg. The posterior end has a spongy-like aeropyle, which is a modification of the exochorion surface (Figure 5D).

The sarcophagid eggs differ from the classical cylindrical shape of those of the Calliphoridae and Muscidae species [22,37,39,40,41,42,43,44,45]. The absence of a hatching line and median area could be related to the fact that the median area has a respiratory function [44], and most Sarcophagidae species are larviparous. The authors Lopes and Leite [43], Sukontason et al. [39], and Pimsler et al. [37] did not observe these structures in the sarcophagid species *P*. (*S*.) *lambens*, *S*. (*L*.) *dux*, and *Blaesoxipha* (*Gigantotheca*) *plinthopyga* (Wiedemann, 1830) either, corroborating this hypothesis. Therefore, embryo respiration must occur only through the spongy-like aeropyle located at the posterior end of the egg, which was also observed by Pimsler et al. [37] for *B*. (*G*.) *plinthopyga*. Based on the description of the Calliphoridae and Muscidae species [22,40,41,42,43,44], the concave surface can be considered dorsal, and the convex surface is considered ventral.

#### 3.2.2. First-Instar Larvae

First-instar larvae have a vermiform format, a pointed anterior region, and a truncated posterior region (Figure 6A). The median length is 1.80 ± 0.26 mm. The larvae body comprises twelve segments: a pseudocephalon, three thoracic segments (TI–TIII), and eight abdominal segments (AI–AVIII). The bilobed pseudocephalon has a pair of antennae composed of a slim distal dome and a basal ring with a lateral receptor pore; the maxillary palps have a simple circular structure around the papillae; and the ventral organs have a single pointed sensillum (Figure 6B). In addition, there is a rudimentary oral ridge with two or three lines starting at the level of the antennae and leading to the oral opening. A cephalic collar surrounds the entire body circumference and is composed of several individuals or grouped flattened filiform spines. The thoracic segments have a complete anterior spine band. The abdominal segments one and two (AI and AII, respectively) have complete anterior spine bands and incomplete posterior spine bands, with the dorsal spines missing. The abdominal segments three through seven have complete anterior and posterior spine bands. The abdominal intersegmental spines have a broad base with thin, hair-like tips (Figure 6C). The anal region has a spiracular cavity. Lateral creeping welts (LCW) are present in the abdominal segments AI through AVII and are not covered with spines. Two spiracular plates with two slits in each can be seen inside the spiracular cavity (Figure 6D). Six pairs of small tubercles can also be seen surrounding the spiracular cavity, each with a papilla at its apex (two inner dorsal tubercles; two median dorsal tubercles; two outer dorsal tubercles; two inner ventral tubercles; two median ventral tubercles; and two outer ventral tubercles).

The shape, body composition, and all the pseudocephalon structures observed in *H. aurescens* were expected for a muscoid species [22,41,42,45,46,47]. However, the observation of the ventral organs in the first-instar larvae seems to be more representative of the sarcophagid species [48,49] than those of the other families, such as Calliphoridae [22] and Muscidae [41,42]. The absence of an anterior spiracular structure is also an expected characteristic for first-instar larvae. In the anal region or abdominal segment AVIII, the spiracular plates inside the spiracular cavity are characteristic of the sarcophagid species [45,47,50,51]. The posterior spiracular openings of *H. aurescens* and other species of its sister groups, *P*. (*P*.) *chrysostoma*, *P*. (*S*.) *lambens*, *Sarcophaga* (*Liopygia*) *argyrostoma* Robineau-Desvoidy, 1830, *Sarcophaga* (*Liopygia*) *cultellata* Pandellé, 1896, and *Sarcophaga* (*Liosarcophaga*) *tibialis* Macquart, 1851, all presented two slits [47,52,53,54,55]. However, this is not sarcophagid-specific [22,42].

#### 3.2.3. Second-Instar Larvae

The second-instar larvae also have a vermiform format with a pointed anterior region and truncated posterior region (Figure 7A). The median length is 3.23 ± 0.43 mm. The divisions of the body segments are as they are in the first instar. However, the pseudocephalon structures are more developed than they are in the previous instar, with a pair of antennae, maxillary palps, and ventral organs. The oral ridges are more complex, starting at the level of the maxillary palps and leading to the oral opening, and the grooves expand to several rows near the oral opening (Figure 7B). The cephalic collar is complete, it surrounds the entire circumference of the larvae body, and it has individual or grouped spines in irregular rows. The spines have a shark-tooth-like shape, and each has a single tip. All the anterior spine bands are complete on the thoracic segments. Anterior spiracles are present on the first thoracic segment (TI), with six or seven ramifications aligned regularly (Figure 7C). Wart-like structures are visible in the middle of the third thoracic segment (TIII). All the anterior spine bands are complete on all the abdominal segments. The posterior spine bands are incomplete since they lack dorsal spines on AI–AIII, but are complete from AIV to AVII. Wart-like structures are also presented on the dorsal surface of the abdominal segments one through seven (AI–AVII). A spiracular cavity in the anal division is surrounded by six pairs of tubercles. Two spiracular plates are visible inside the cavity with two linear openings (Figure 7D).

The form of the cephalic collar spines of *H. aurescens* is like the ones observed in *Peckia* (*Euboettcheria*) *collusor* (Curran & Walley, 1934) and *P*. (*S*.) *lambens*, which is the closest genus to the one studied here [45,47]. The presence of one tip on each spine of the cephalic collar differs from the multiple tips found on *Ravinia belforti* (Prado e Fonseca, 1932) [50]. When comparing the anterior spiracle of sarcophagid species, *H. aurescens* shows fewer ramifications than the other Sarcophaga-clade species [14], and it is the species with the lowest number, followed by *S*. (*L*.) *ruficornis* (8–12 ramifications), *S*. (*L*.) *argyrostoma* (11–12 ramifications), *P*. (*S*.) *lambens* (11–13 ramifications), *P*. (*E*.) *collusor* (11–14 ramifications), *S*. (*L*.) *cultellata* (15 or 16 ramifications), and *S*. (*L*.) *tibialis* (15–20 ramifications) [45,47,52,54,55,56]. *Oxysarcodexia paulistanensis* (Mattos, 1919) and *R. belforti* show an increased number, with 14 and 16–22 ramifications, respectively [50,53]. Furthermore, the spiracle ramifications present a linear organization similar to the conformity observed in *S*. (*L*.) *argyrostoma, S*. (*L*.) *cultellata*, and two species of the subfamily Paramacronychiinae [52,55,57], although they differ from the irregular organization of *R. belforti*, *O. paulistanensis*, and *S*. (*L*.) *tibialis*, being the last with an irregularity organization in the middle of the anterior spiracle ramifications [50,53,54].

The presence of developing tubercles surrounding the spiracular cavity in the second-instar larvae is more frequently observed in Sarcophagid species than it is in the other families, but it does not make this an exclusive characteristic [22,41,45,47,50,52,53]. The spiracular plates in all the second instar sarcophagid larvae are still located inside a cavity and are described as having two straight openings.

#### 3.2.4. Third-Instar Larvae

The third-instar larvae have a vermiform format with a pointed anterior and truncated posterior region. The median length is 6.53 ± 0.54 mm. The pseudocephalon has all the completely developed structures, with a domed shape antenna, maxillary palps with semi-disc structures surrounding the papillae, oral ridges composed of several grooves, and a pair of ventral organs between the maxillary palps and the oral ridges (Figure 8A,B). The cephalic collar or spines between the pseudocephalon and the first thoracic segment have at least eight complete rows (Figure 8C). The spines are located individually or in groups of up to four, and with one or two tips. A pair of anterior spiracles with eight ramifications in a regular row is located dorsolaterally in the posterior region of TI. The intersegmental spines (TIII–AI) are more concentrated on the dorsal surface than they are on the lateral and ventral surfaces; the rows are seemingly interrupted on the lateral sides at the level of the anterior spiracles. The spines have a shark-tooth-like shape, are organized singly or in pairs, and have a single or double tip. The papillae are organized in a row in the middle of the segments AI–AVII. The integument of the abdominal segments (AI–AVIII) is covered with wart-like structures. The dorsal and lateral posterior spines were only present on the AVI and AVII segments and face anteriorly. No anterior dorsal spines are present on the AVIII segment. The anal division or AVIII has a deep cavity where the posterior spiracles are located. Spiracular plates have three linear slits. The presence of a spiracular scar could not be seen. The spiracular cavity is surrounded by six pairs of tubercles and filiform spines, which face out of the cavity (Figure 8D,E). The ventral surface of the AVIII segment is covered with small spines compared to the intersegmental ones. The perianal pad that has an anal opening, and two anal tubercles are covered with spines, and the anal papillae are located at the apex of the tubercles (Figure 8F).

The number of ramifications on the anterior spiracle of *H. aurescens* is one of the lowest among the studied species in the “Sarcophaga-clade”. The anterior spiracles of *H. aurescens* have eight ramifications that are similar to those of *Sarcophaga* (*Helicophagella*) *hirticrus* Pandellé, 1896, which has from seven to ten, and *Sarcophaga* (*Liosarcophaga*) *exuberans* Pandellé, 1896, with ramifications varying from eight to eleven. In sequence, they are *Sarcophaga* (*Liosarcophaga*) *nodosa* Engel, 1925, *S*. (*L*.) *argyrostoma*, *S*. (*B*.) *africa*, *P*. (*S*.) *lambens*, *P*. (*E*.) *collusor*, *Sarcophaga* (*Heteronychia*) *javita* (Peris, González-Mora & Mingo-Pérez, 1998), *S.* (*L.*) *tibialis*, *Peckia* (*Euboettcheria*) *anguilla* (Curran & Walley, 1934), *P*. (*P*.) *intermutans*, and *P*. (*P*.) *chrysostoma* [25,45,47,50,52,54,56,58,59]. Even when *H. aurescens* is compared to *H. pilipleura* (13–15 ramifications), the number of ramifications on the anterior spiracle is lower [15], suggesting that this structure could be useful to distinguish between both species. 

In addition to the number of ramifications, their organization has some importance in species differentiation for all the species previously referred to. Only *S*. (*L*.) *tibialis* and *P*. (*P*.) *chrysostoma* have the irregular disposal of the anterior spiracular orientation, and the others show a regular organization, which is similar to what is normally seen in Muscidae and Calliphoridae specimens and also in *H. pilipleura* [15,22,41,42,54,60]. Six pairs of tubercles surrounding the spiracular cavity are still present in the third-instar larvae, which is the same as all the other sarcophagid species belonging to the “Sarcophaga-clade”, including the one in this study, *H. aurescens*. Furthermore, all of them have an incomplete peritreme on both posterior spiracular plates, as in *H. pilipleura* [15,25,45,47,50,56,58,59].

#### 3.2.5. Puparia

The puparia have a cylindrical format, and the anterior spiracles at the posterior part of TI are the most anterior part of this stage (Figure 9A). The pseudocephalon and the anterior part of TI are internalized (Figure 9B). Wrinkles are visible all over the remaining segments due to the contraction of the body for the formation of the puparium (Figure 9C). The median length is 5.40 ± 0.17 mm. The abdominal intersegmental spines have kept their shark-teeth-like shape. Wart-like structures were also observed all over the abdominal segments. The spiracular cavity has a spiracular plate with three linear slits, and spiracular scars are visible on each plate (Figure 9D,E). The opposite side of the spiracular plates is covered with several groups of one-tip spines of different sizes (Figure 9F). Six pairs of tubercles surrounding the spiracular cavity are still present. The tissue between the tubercles and the spiracular cavity has groups of one to seven hair-like spines all over the surface.

SEM descriptions of the sarcophagid puparia are included in fewer publications than those on the other immature stages. However, the puparium morphology is very similar to that of the third-instar larvae. During the process, the pseudocephalon and a part of the first thoracic segment are invaginated, making the anterior spiracles the most anterior part of this stage. For the species *P*. (*E*.) *collusor*, the number of anterior spiracle ramifications varies from 15 to 16, while *S*. (*L*.) *dux* has from 14 to 17, showing a higher number of ramifications than those of *H. aurescens* [45,50]. Besides the retraction of the first segments, the entire body contracted, making the puparium surface wrinkled, but not preventing the observation of the structures described in the L3-instar larvae. The posterior spiracles maintain the three openings and an incomplete peritreme in all the species previously mentioned [45,54,55,61].

## 4. Conclusions

This study contributes to the knowledge of the immature morphology and the post-embryonic development of the Sarcophagidae family and Neotropical sarcophagid fauna. 

The increase in temperature caused an acceleration in the development time of *H. aurescens*, as well as a decrease in the larval weight. The viability in the period from the newly hatched larvae (L1) phase to the adult period was also lower at this temperature, which may indicate that the viability threshold is close to the temperature of 29 °C. 

The morphological analysis allowed us to describe structures, such as the number and arrangement of spiracle openings, which can help in the identification of *H. aurescens*, even when compared with another of the same genus. The combination of structures presented and the bionomics aspects of *H. aurescens* could be useful to differentiate this species among the others belonging to the Sarcophagidae family.

## Figures and Tables

**Figure 1 insects-15-00753-f001:**
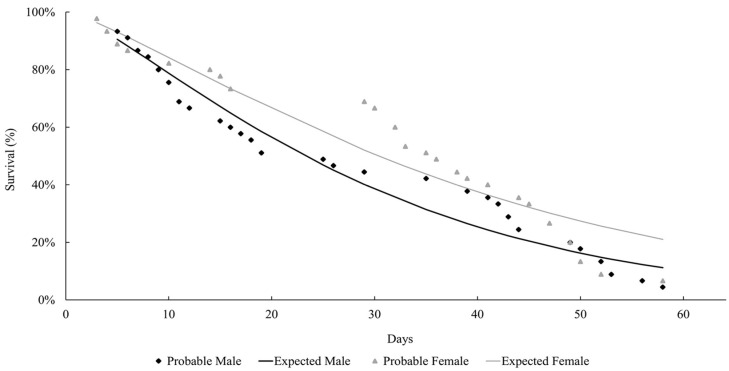
Longevity of adults of *Helicobia aurescens* (Diptera: Sarcophagidae: Sarcophaginae) at the temperature of 27 ± 1 °C.

**Figure 2 insects-15-00753-f002:**
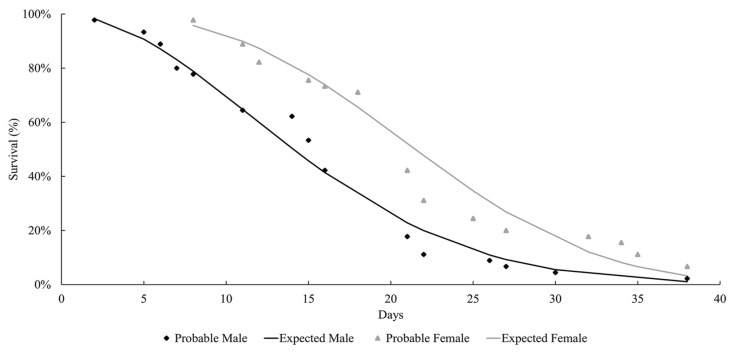
Longevity of adults of *Helicobia aurescens* (Diptera: Sarcophagidae: Sarcophaginae) at the temperature of 29 ± 1 °C.

**Figure 3 insects-15-00753-f003:**
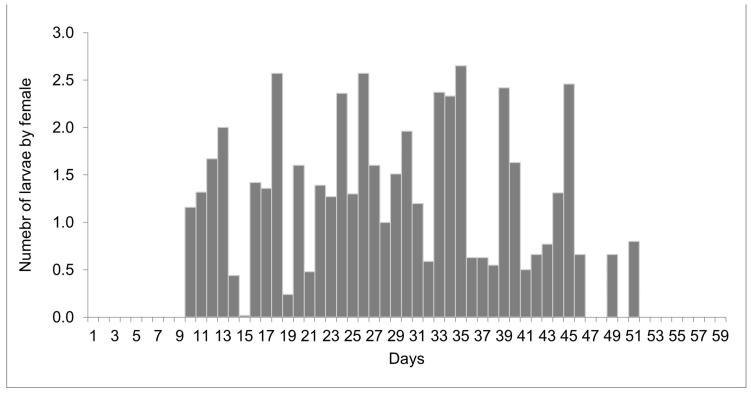
Number of larvae laid by females of *Helicobia aurescens* (Diptera: Sarcophagidae: Sarcophaginae) at the temperature of 27 ± 1 °C.

**Figure 4 insects-15-00753-f004:**
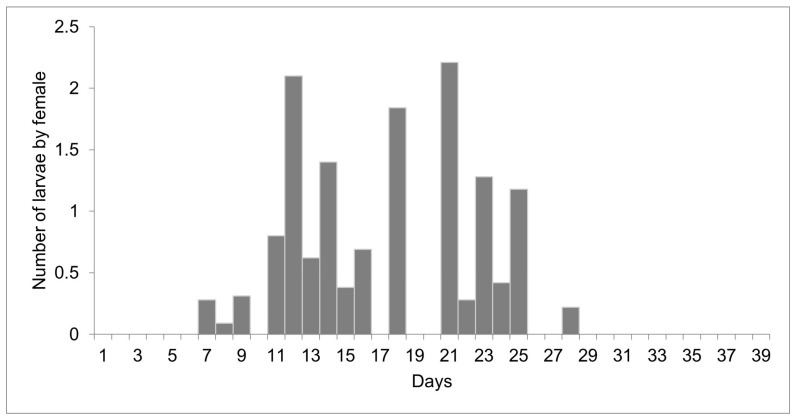
Number of larvae laid by females of *Helicobia aurescens* (Diptera: Sarcophagidae: Sarcophaginae) at the temperature of 29 ± 1 °C.

**Figure 5 insects-15-00753-f005:**
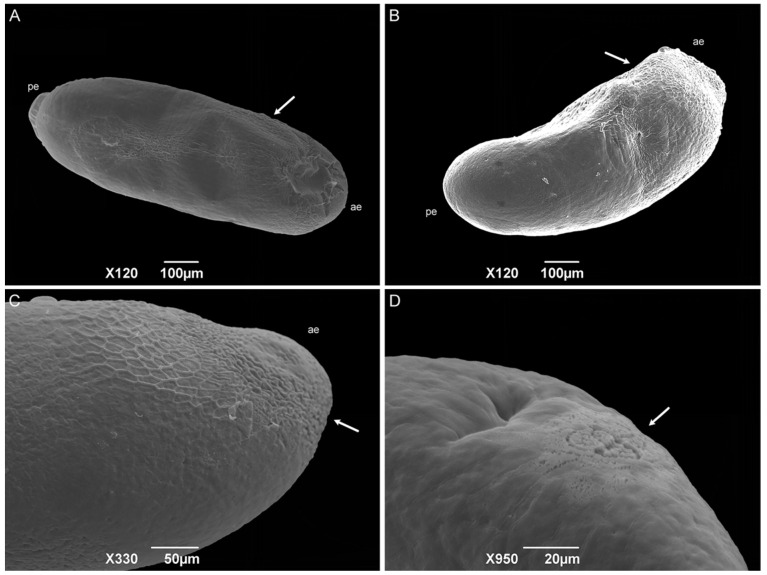
Scanning electron microscopy of *Helicobia aurescens* (Diptera: Sarcophagidae) eggs. (**A**,**B**) Egg showing the anterior end (ae), posterior end (pe), and the hexagonal pattern of the chorionic cells (arrow). (**C**) Anterior end (ae) with a more detailed observation of the hexagonal pattern (arrow). (**D**) Aeropyles (arrow) located at the posterior regions. Pictures: Electron Microscopy Platform Rudolf Barth—IOC/FIOCRUZ.

**Figure 6 insects-15-00753-f006:**
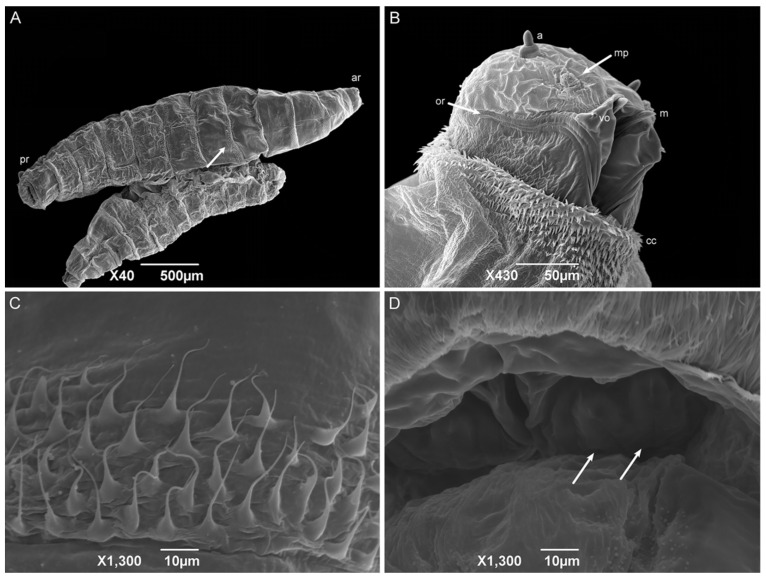
Scanning electron microscopy of *Helicobia aurescens* (Diptera: Sarcophagidae) first-instar larvae. (**A**) Larval body with the anterior region (ar), posterior region (pr), and highlighting of the intersegmental spines (arrow); (**B**) Ventral–lateral view of the pseudocephalon with an antenna (a), maxillary palps (mp), the ventral organ (vo), the oral ridges (or), the cephalic collar (cc), and the mandible (m). (**C**) Intersegmental spines composition and characteristics. (**D**) Posterior spiracle with two openings (arrows) located inside a spiracular cavity. Pictures: Electron Microscopy Platform Rudolf Barth—IOC/FIOCRUZ.

**Figure 7 insects-15-00753-f007:**
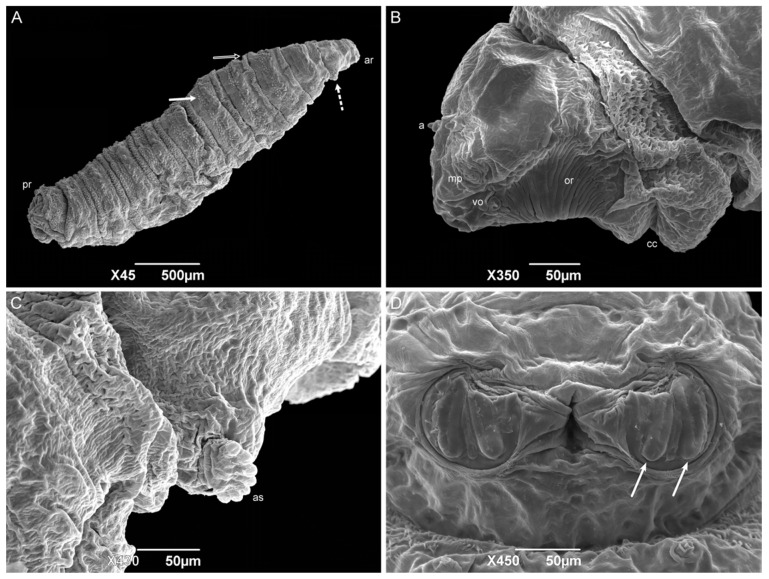
Scanning electron microscopy of *Helicobia aurescens* (Diptera: Sarcophagidae) second-instar larvae. (**A**) Larval body with the anterior region (ar), followed by the anterior spiracle (dot arrow), the intersegmental spines (white arrow and black arrow), and the posterior region. (**B**) Lateral view of the pseudocephalo with antennae (a), maxillary palps (mp), ventral organs (vo), oral ridges (or), and the cephalic collar (cc). (**C**) Detail of first thoracic segment showing the right anterior spiracle (as) with seven ramifications. (**D**) Posterior spiracle with two openings (arrow). Pictures: Electron Microscopy Platform Rudolf Barth—IOC/FIOCRUZ.

**Figure 8 insects-15-00753-f008:**
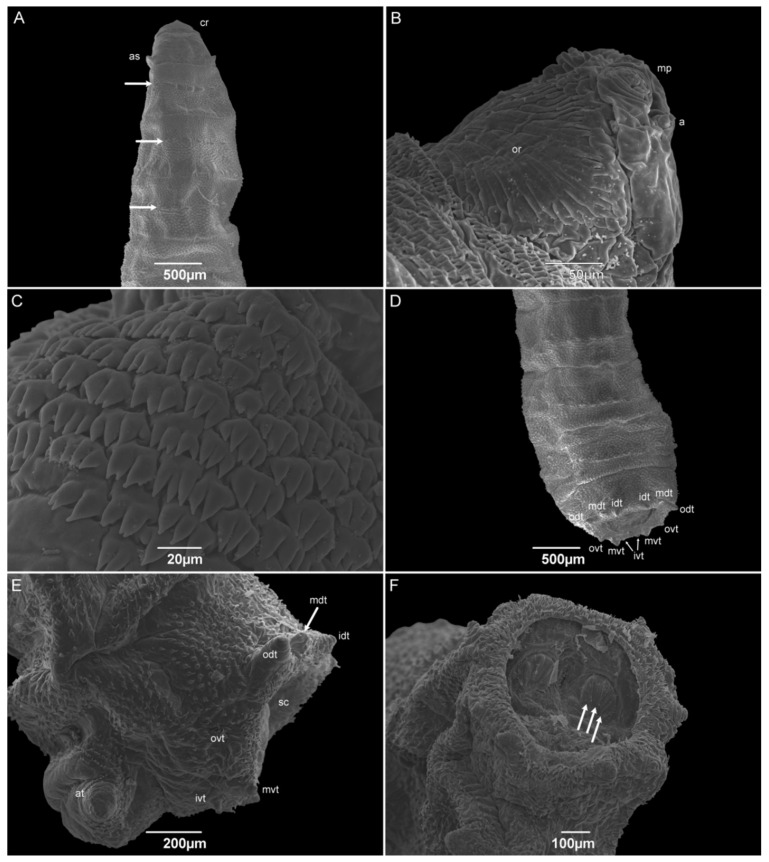
Scanning electron microscopy of *Helicobia aurescens* (Diptera: Sarcophagidae) third-instar larvae. (**A**) Anterior region highlighting the cephalic region, anterior spiracle (as), and wart-like structures (arrow) in the middle of thoracic segments II and III and first abdominal segment. (**B**) Details of the pseudocephalo showing the antennae (a), maxillary palps (mp), and the oral ridges (or). (**C**) Details of the spines of the cephalic collar in a dorsal view. (**D**) Posterior regions showing the eight tubercles surrounding the spiracular cavity. Two pairs of outer dorsal tubercle (odt), medium dorsal tubercle (mdt), inner dorsal tubercle (idt), outer ventral tubercle (ovt), medium ventral tubercle (mvt), and inner ventral tubercle (ivt and arrow). (**E**) Lateral view of the anal region showing the left outer, medium, and inner dorsal and ventral tubercles (odt, mdt and arrow, idt, ovt, mvt, and ivt), the spiracular cavity (sc) and the anal tubercle (at). (**F**) Posterior view of the anal region showing two spiracular plates with three openings (white arrows). Pictures: Electron Microscopy Platform Rudolf Barth—IOC/FIOCRUZ.

**Figure 9 insects-15-00753-f009:**
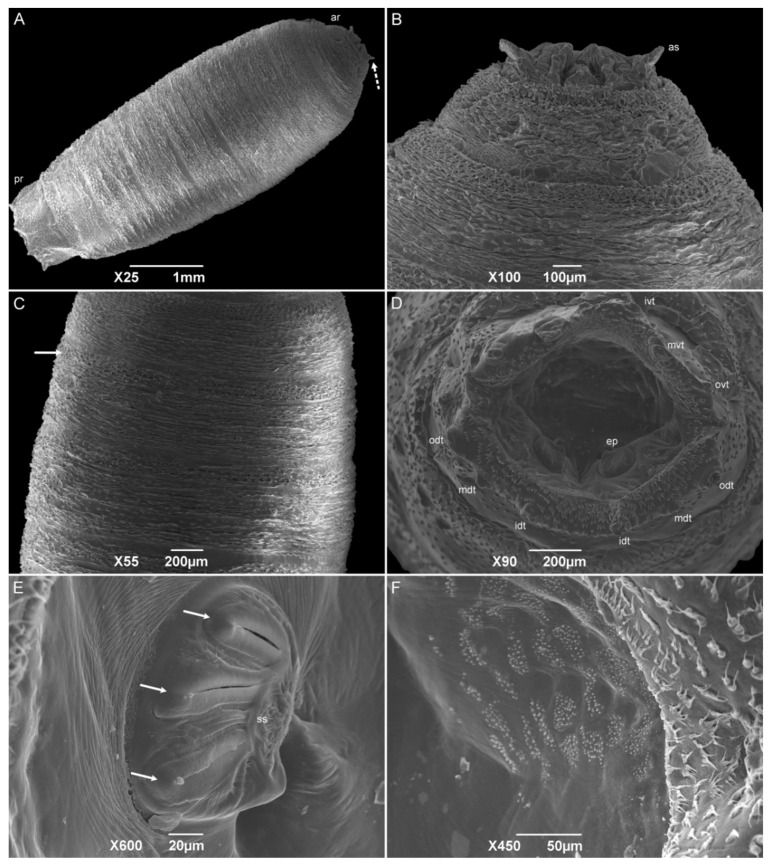
Scanning electron microscopy of *Helicobia aurescens* (Diptera: Sarcophagidae) puparia. (**A**) Puparium body showing the anterior region with the anterior spiracle (as and dotted arrow) and posterior region. (**B**) Details of the anterior regions showing the anterior spiracles. (**C**) Central part of the puparium showing the intersegmental spines (arrow). (**D**) Posterior view of the anal region showing the spiracular cavity with two spiracular plates and surrounded by the tubercles (odt, mdt, idt, ovt, mvt, and ivt). (**E**) Details of the left posterior spiracular plate with three openings (white arrow) and a spiracular scar (ss). (**F**) Details of the opposite side of the spiracular plates with spines all over the surface. Pictures: Electron Microscopy Platform Rudolf Barth—IOC/FIOCRUZ.

**Table 1 insects-15-00753-t001:** Duration of post-embryonic development and viability of *Helicobia aurescens* (Diptera: Sarcophagidae: Sarcophaginae) reared at two different temperatures.

Biological Features	Temperature: 27 ± 1 °C				Temperature: 29 ± 1 °C			
	Duration (Days)		Viability		Duration (Days)		Viability	
	Mean ± SD	Range	%	*n*	Mean ± SD	Range	%	*n*
Larval Stage	5.37 ± 0.75 ^a^	4–8	96	191	4.83 ± 0.86 ^b^	3–9	82	164
Pupal Stage	10.38 ± 0.59 ^a^	9–11	89	170	8.32 ± 0.61 ^b^	6–10	82	134
Newly Hatched (L1) Larvae to Adult Period	15.62 ± 0.66 ^a^	14–17	85	170	13.06 ± 0.51 ^b^	12–15	67	134

Abbreviations: SD—standard deviation. The lowercase letters represent there is no statistical difference between the temperatures and developmental periods.

**Table 2 insects-15-00753-t002:** Mature larval weight (mg) of *Helicobia aurescens* (Diptera: Sarcophagidae: Sarcophaginae) reared at two different temperatures.

Biological Parameter/ Development Stage	Temperature: 27 ± 1 °C		Temperature: 29 ± 1 °C	
	Mean ± SD	Range	Mean ± SD	Range
Females	16.80 ± 1.82 ^a^	14–21	14.72 ± 1.81 ^b^	11–19
Males	15.21 ± 1.45 ^a^	12–20	13.00 ± 1.62 ^b^	7–17
Females and Males	16.19 ± 2.01 ^a^	12–22	13.57 ± 2.38 ^b^	5–20
Abandon on the 3rd Day	16.64 ± 1.79 ^a^	14–21	14.34 ± 2.03 ^b^	11–20
Abandon on the 4th Day	15.94 ± 2.08 ^a^	12–22	13.06 ± 1.65 ^b^	9–18
Abandon on the 5th Day	17.00 ± 2.82 ^a^	15–19	11.66 ± 3.28 ^c^	5–16

Abbreviations: SD—standard deviation. The lowercase letters represent there is no statistical difference between the temperatures and developmental periods.

## Data Availability

All relevant data are included within the paper.

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
