# Peer review of "Ultrastructure of Immatures Stages and Life Cycle of Helicobia aurescens (Diptera: Sarcophagidae: Sarcophaginae)"

_insects, 2024, doi:10.3390/insects15100753_

Round 1

Reviewer 1 Report

Comments and Suggestions for Authors

Review – Cortinhas, et al. Ultrastructure of immature stages and life cycle of Helicobia aurescens (Diptera: Sarcophagidae: Sarcophaginae).

This paper provides new information on the life history of Helicobia aurescens and the morphology of its immature stages. The descriptions of the external anatomy of the egg, larvae, and puparium are pretty well written, but there are a number of problems with grammar and writing style in the opening introduction and discussion.

A couple of general problems that showed up throughout the manuscript are:

·       Citations – sometimes just a number and sometimes with author, date and number.  I am used to author, date citations but check with editor on what they want. This could also affect how you arrange your references at the end of the paper.

·       Do not need to include (Diptera:Sarcophagidae) after each mention of a species name (except in title).  This can be assumed.  If you are making comparisons to other families then it would be good to note that.  Otherwise you can assume your readers know that Helicobia, Sarcophaga, Ravinia, etc. are sarcophagids.

·       There should be consistency in abbreviation of genus and subgenus names after first mention.  Sometimes you abbreviate P. for Peckia, other times you write it out.

·       Author names should be included in the species name on first mention of that species and do not need to be included after that.

·       Paragraphs should be longer than one sentence.  Combine such short paragraphs with others.

A couple of general comments and suggestions that might make this paper more useful, if you have the specimens available:

·       One article you did not mention, and it is the first article I would suggest for a paper on sarcophagid larvae, is: Sanjean, J. 1957. Taxonomic studies of Sarcophaga larvae of New York, with notes on the adults. This has a nice description of H. rapax larvae that might be useful to compare with.

·       For forensic work the length of the larval instars is important to know, along with the temperature for calculation of PMI.  You have weight information, but nothing on length of larvae (killed with hot water).

·       You have nice descriptions and SEM photos of the larval instars, but no depictions of the cephalopharyngeal skeletons. 

·       Line drawings of the posterior end of the larvae showing the size and location of the tubercles and spiracular plates would be useful.  This is what is often used to help with identification of larvae. It is hard to see this in the SEM photos.  (See: https://www.cdc.gov/nceh/ehs/docs/pictorial_keys/flies.pdf )

More detailed comments:

Simple Summary

Line 13: do not capitalize forensic

Line 15: delete “So the aim was to …” Change to We describe aspects of the life cycle …”

Line 22: puparia is plural, so use “are” instead of “is”

Abstract.

Line 24-25: Most general statements should not be included in the abstract. They might be OK in the introduction.  Delete “However, as there … morphology”. Start sentence with This study aims to …

Line 27: change ‘most’ to ‘greatly’?

Line 28: delete end of sentence “…which directly affects…”

Line 32: change “sarcophagids” to “sarcophagid”

Line 32: delete “However, on observing” – start sentence with “The second instar has the anterior spiracles present …”

Line 36-38: delete last sentence.  Too general for abstract.

Introduction

Line 48: change “has” to “inhabit”

Line 53-54: A one sentence paragraph.  Change wording and include in one of your other introductory paragraphs.

Line 55: need to reword opening sentence.

Line 56: delete “However, usually” add “usually” between “are” and “found”

Line 58-59: delete last sentence

Line 60-63: I suggest expanding this paragraph to provide summary of known biology of this species and other members of Helicobia.

Line 67-70: I would delete this paragraph or rewrite to summarize problems with taxonomic position over time.

Materials and Methods

I really do not understand what you are trying to say in the 2.1 Fly rearing paragraphs.  Are these flies originally from an old culture started in 2009 by Barbosa et al.? Use this part of your paper to explain where the flies came from, how many generations have they gone through in the laboratory, and what your general rearing and maintenance procedures are.

Line 161: I do not understand this sentence.  How are calliphorids (with small c) involved?

Line 150: Change “…difference regarding …” to “when the larvae abandoned ..”  I use abandoned here to match the wording in Table 2.

      Note: I do not really understand how larvae who fed for a shorter time weighed more than larvae that fed for a longer time.

Line 165: change “light” to “low”

Line 178 – Table 2: Does this really say that larvae only fed for 1, 2, or 3 days before abandonment? This does not make sense to me.  How are they going through full development in a single day?

Line 263: Are you indicating that the first instar larva has two pairs of antennae? This is not indicated in Figure 4B.

For your descriptions of the egg, larval instars, and puparium, you do not indicate how many specimens the descriptions are based on.

Comments on the Quality of English Language

Some trouble with grammar, especially in introductory materials.  Better when it came to the descriptions.

Author Response

Manuscript ID: insects-2673260

Title: Ultrastructure of immatures stages and life cycle of Helicobia aurescens (Diptera: Sarcophagidae: Sarcophaginae).

Authors: Lucas Barbosa Cortinhas, Paloma Martins Mendonça*, Eliane Gomes Perrut, Rodrigo Rocha Barbosa, Jacenir Reis dos Santos-Mallet, Margareth Maria de Carvalho Queiroz

All suggestions and corrections were made directly on the manuscript (marked in red). The Introduction, Methodology, and part of the Results and Discussion were rewritten, according to the reviewers’ suggestion.

Figures 1 and 2 were separated and included in the manuscript with a better resolution. All figures had to be renumbered. 

Regarding the discussion of the immature description part, the authors agreed that since we described the stages using SEM, the discussion should be with papers using the same technique.

The authors decided to keep the biology and morphology of H. aurescens in the same manuscript, understanding that, as with other sarcophagids, it is important to know the biological aspects of this species. Data on biological aspects can serve as a basis for the development of more detailed studies, working more effectively with forensic entomology.

We hope that could answer all the suggestions made by the reviewers.

Reviewer 2 Report

Comments and Suggestions for Authors

Major Comments

Though there is valuable data presented in this manuscript, it is so poorly written that it is impossible to get through. Even when the development data is presented, the figures cannot be read! This manuscript must go through a major revision to be acceptable for publishing. I gave some suggestions on re-phrasing, but I had to stop halfway through because I refuse to re-write the entire paper for the authors.

The paper is also written so that almost every sentence is its own paragraph. This is completely unacceptable and demonstrates that the authors are unfamiliar with the journal submission guidelines.

The title is also poorly written and confusing. It should be revised to something along the lines of “Description of immature development and ultrastructure of Helicobia aurescens…”

One major change I recommend to the authors is to only present the SEM data. The development data takes away from the interesting part of the paper, which is the ultrastructural morphology. The SEM figures are nice and there seems to be WAY more information on this part of the project anyway. The development data should be submitted as its own technical note elsewhere.

Minor Comments

L. 13 – “Forensic” does not need to be capitalized.

L. 14 – Change “of post-morten interval" to “of the postmortem interval”.

L. 15 – Change “specie” to “species”.

L. 15 – Remove “So” at the beginning of the sentence.

L. 17 – Change “fleshfly” to “flesh fly”.

L. 17 – Consider re-phrasing to “Lower temperatures increased total development time”. Revise similarly throughout.

L. 18 – Change “were” to “was”. Revise similarly throughout.

L. 19 – Is “eggshell” the correct term here? Or is “chorion” more appropriate?

L. 23 – Change “as there is” to “as there are”. Revise similarly throughout.

Introduction – Why are there so many short paragraphs? Many of them only have a single sentence, which is not acceptable. Please consolidate into one or two paragraphs. For example, L. 44 – 66 could be one paragraph, and L. 68 – 73 could be the second paragraph.

Throughout the remainder of the paper – please consolidate your paragraphs! A paragraph should have no less than three sentences, minimum.

L. 121 – Remove “As can be seen in Table 1” and just put Table 1 in parentheses: “…lasts longer at 27C than at 29C (Table 1).” Revise similarly throughout.

Tables 1 and 2: Change “Biological Parameter” to “Development Stage”.

L. 147 – Revise to “Sex had no significant impact on larval weight for either temperature (P > 0.05; Table 2). Revise similarly throughout.

Figures 1 and 2: Your axes are impossible to read! Stack the plots vertically instead of horizontally and the increase your font size.

Conclusions??

There is no summary paragraph!

Comments on the Quality of English Language

Poor. Needs revision.

Author Response

Manuscript ID: insects-2673260

Title: Ultrastructure of immatures stages and life cycle of Helicobia aurescens (Diptera: Sarcophagidae: Sarcophaginae).

Authors: Lucas Barbosa Cortinhas, Paloma Martins Mendonça*, Eliane Gomes Perrut, Rodrigo Rocha Barbosa, Jacenir Reis dos Santos-Mallet, Margareth Maria de Carvalho Queiroz

All suggestions and corrections were made directly on the manuscript (marked in red). The Introduction, Methodology, and part of the Results and Discussion were rewritten, according to the reviewers’ suggestion.

Figures 1 and 2 were separated and included in the manuscript with a better resolution. All figures had to be renumbered. 

Regarding the discussion of the immature description part, the authors agreed that since we described the stages using SEM, the discussion should be with papers using the same technique.

The authors decided to keep the biology and morphology of H. aurescens in the same manuscript, understanding that, as with other sarcophagids, it is important to know the biological aspects of this species. Data on biological aspects can serve as a basis for the development of more detailed studies, working more effectively with forensic entomology.

We hope that could answer all the suggestions made by the reviewers.

Paloma Martins Mendonça

Reviewer 3 Report

Comments and Suggestions for Authors

This is a interesting contribution to the knowledge of species of forensic interest. Despite of it, the ultrastructural study suffers from lack of an optical microscopy study since it is known that several features are best seen with this methodology than with SEM or, as in the case of the cephalopharyngeal skeleton, only visible with optical microscopy. Both techniques have shown to be complementary and useful for studying larval micromorphology (e.g., Ubero-Pascal et al. 2010 (In: Méndez-Vilas A, Díaz J (eds) Microscopy: Science, Technology, Applications and Education. Formatex, Badajoz, pp 1548–1556) or Szpila and Villet 2011 ( J Med Entomol 48:738–752)).

Furthermore, there is literature not considered in the work, such as, in example, that related to Sarcophaga cultellata (Ubero-Pascal et al., 2015. Microscopy Research and Technique, 78: 148-172) or Sarcophaga tibialis (Paños-Nicolás et al., 2015, Pasaitol. Res., 114: 4031-4050), which is of potential interest to compare with the data provided. Moreover, no mention to Grella (20169 is done in the Introduction, despite his study on several species of Sarcophagidae.

The Material and Methods section should be revised and improved since it is unclear  how many individuals were treated for the life cycle study, how many replicates were done, where the individuals for SEM came from... as well as the trade mark, model, etc of the different apparatus used for weighing the larvae and other procedures. Why such two temperatures were selected? Why 25ºC was not considered? The only data concerning development of Helicobia aurescens were got at 25ºC; then, a direct comparison could be made if this temperature was considered.

The Results and Discussion section should be improved by considering more literature to compare. In my opinion, a discussion should not be just a list of the findings of others but rather process the information of the literature more. On the other hand, some paragraphs seem more appropriate for Introduction. The whole drafting should be revised and improved.

Concerning the life cycle presented, it draws attention that no data are provided related to the duration of the different larval stages but only on the total larval duration. If the forensic interest of the study is claimed, the duration of each stage would be needed to can estimate the minimum postmortem interval. 

As regards the description of the different larval stages, comparison with other species (i.e., S. cultellata or S. tibialis) is missing. In the case of the third instar larva, a comparison with data due to Grella (2016) is also missing. The same for puparia.

I'll try to point, in the manuscript, more comments.

Author Response

(The authors gave the same response as above.)

Round 2

Reviewer 2 Report

Comments and Suggestions for Authors

Thank you for this revision. I have no issues with the manuscript as is.

Author Response

Thanks for accepting all suggestions done.

Reviewer 3 Report

Comments and Suggestions for Authors

The new version is quite like the previous one. Despite the authors state that several sections have been rewritten, the fact is that most of them are almost identical than before. Few changes can be seen, at least substantial changes. Concerning the general drawing, there are still unconnected phrases, like ideas that have not been developed, mainly when referring data form the literature.

The authors insist in the interest of knowing the accurate development time of each species to be applied to the PMI estimation, but the research design does not allow such knowledge because the data provided do not distinguish the different larval stages.

In my opinion, the Introduction section does not adequately inform about the background and interest of the study to be addressed.

The Materials and Methods section is very similar to the previous one. The Fly rearing point is almost identical; only the last phrase has been modified although all the point 2.1 appears in red. Other points do not answer some aspects suggested by me, such as trademark and model of the different apparatus used. No mention to the number of replicates, if any, were performed. If two temperatures were considered, how many larvae were used for each one? (200 larvae: 4 groups of 50 individuals… how many per temperature?).

Concerning Results and Discussion, I miss the N parameter in the tables. If a statistical analysis has been applied, N should be known. Table 1 remains a bit unclear. What is the difference between newly-hatched larvae to adult period and Larval stage and Pupal stage?

At least one phrase, modified from the previous version, includes an erroneous cite. Lines 151-153 relate flesh-flies and reference [31], but such paper does not mention flesh-flies or Sarcophagidae at all.

Figures 3 and 4 are located after phrases where eggs are mentioned, but nothing about eggs appears in the figures.

Third instar larvae morphology should be compared with that of Grella (2016) although this author worked with optical microscopy of other species of the genus Helicobia and optical and SEM microscopy for other Sarcophagidae genera. Comparison with mainly H. pilipleura, species of the same genus that the species studied, should be very interesting, in my opinion more interesting (or alike) than a comparison with species of Sarcophaga or Peckia genera.

The Conclusion section includes two phrases the first of which would be better at the Introduction and is not a real conclusion.

More comments can be found in the revised file.

Author Response

We hope that could answer all the suggestions made by the reviewers.

Round 3

Reviewer 3 Report

Comments and Suggestions for Authors

The manuscript has been improved. Nevertheless, Tables should be improved by adding the N for each case. Other minor details are also indicated in the file.

Author Response

All suggestions were accepted. 
